# Digital Smile Designed Computer-Aided Surgery versus Traditional Workflow in “All on Four” Rehabilitations: A Randomized Clinical Trial with 4-Years Follow-Up

**DOI:** 10.3390/ijerph18073449

**Published:** 2021-03-26

**Authors:** Francesca Cattoni, Luca Chirico, Alberto Merlone, Michele Manacorda, Raffaele Vinci, Enrico Felice Gherlone

**Affiliations:** Department of Dentistry, IRCCS San Raffaele Hospital, 20132 Milan, Italy; cattonif@tiscali.it (F.C.); albertomerlone@tiscali.it (A.M.); micmanacorda@gmail.com (M.M.); raffaelevinci@libero.it (R.V.); gherlone.enrico@hsr.it (E.F.G.)

**Keywords:** dental implant, digital dentistry, full-arch rehabilitations, All on Four, implant survival, implant-prosthodontic restorations

## Abstract

The aim of the present study was to evaluate and compare the traditional “All on Four” technique with digital smile designed computer-aided “All on Four” rehabilitation; with a 4-years follow-up. The protocol was applied to a total of 50 patients randomly recruited and divided in two groups. Digital protocol allows for a completely virtual planning of the exact position of the fixtures, which allows one to perform a flapless surgery procedure with great accuracy (mini-invasive surgery) and also it is possible to use virtually planned prostheses realized with Computer-Aided Design/Computer-Aided Manufacturing (CAD/CAM) (methods for an immediate loading of the implants. After 4 years from the treatments 98% of success were obtained for the group of patients treated with the traditional protocol and 100% for the digital protocol. At each time interval a significant difference in peri-implant crestal bone loss between the two groups was detected; with an average Marginal Bone Loss (MBL) at 4 years of 1.12 ± 0.26 mm in the traditional group and 0.83 ± 0.11 mm in the digital group. Patients belonging to the digital group have judged the immediate loading (92%), digital smile preview (93%), the mock-up test (98%) and guided surgery (94%) as very effective. All patients treated with a digital method reported lower values of during-surgery and post-surgery pain compared to patients rehabilitated using traditional treatment. In conclusion, the totally digital protocol described in the present study represents a valid therapeutic alternative to the traditional “All on Four” protocol for implant-supported rehabilitations of edentulous dental arches.

## 1. Introduction

The therapeutic efficacy of rehabilitations based on the use of a reduced number of implants, with a high aesthetic and functional yield is now universally recognized [1,2,3,4]. Among the most adopted implantology protocols for the treatment of dental arches with moderate/severe bone atrophy, the “All on Four” technique continues to achieve great success among the scientific community [5,6,7,8]. This method involves the placement of four implants: two axial ones positioned in the anterior sector and two inclined at about 30–35° with respect to the occlusal plane in the lateral alveolar areas. This inclination allows it to distalize the implant emergency and to provide support to a prosthetic arch up to the first molar, and to avoid any damage the noble structures such as the maxillary sinus (upper arch) and the inferior alveolar vascular-nerve bundle (lower arch). This also prevents the bone regeneration procedure in the presence of severe atrophies [5]. In recent years, digital technologies have significantly changed the clinical dental practice with regards to diagnosis, prosthetic planning, guided surgery and implant-supported rehabilitations [9,10,11]. With the recent introduction of software specifically programmed for clinicians and dental technicians, it is possible to combine the aforementioned procedures [12]. It is therefore possible to elaborate an implant-prosthetic rehabilitation, even in the more complex scenarios, through the use of a software, thus having the opportunity to previsualize the final result and consequently improve the communication between the clinician and the patient, and between the prosthodontist, surgeon and dental technician, also achieving a better quality of the project and the final result [13,14].

From 2014, clinicians of the Department of Dentistry of the Vita-Salute San Raffaele University have developed and applied a specific digital protocol, which involves digital implant-prosthetic planning, flapless-guided surgery and digital impression. Additionally, more than 4 years later, the results of this work highlighted the main differences between a digital and a traditional method.

The aim of the present study is therefore to describe and assess the two protocols on two homogeneous groups of patients, evaluating:Marginal bone level values (at 12, 24, 36 and 48 months) by radiographic evaluation;Implant and prosthetic complications and failures;Appreciation by the patient of the procedures used;Evaluation of operative and post-operative pain.

## 2. Materials and Methods

### 2.1. Patients Selection

The implant-prosthetic protocol was conducted including a population of 50 patients aged between 46 and 85, who underwent rehabilitation of the edentulous maxilla with a reduced number of implants, At the Department of Dentistry (San Raffaele-Milan), directed by Prof. E. F. Gherlone.

Twenty-five patients were randomly selected and subjected to the implant-prosthetic protocol with the digital method. The remaining twenty-five underwent the traditional “All on Four” protocol (Figure A5).

Inclusion criteria were: patients of any ethnicity over 18 years of age, male and female; patients with good general health, without chronic disease (immunosuppression, untreated coagulation problems, chemotherapy and radiotherapy, assumption of bisphosphonate drugs, cardiac conditions and uncompensated diabetes). The selected patient must have had at least one totally edentulous arch or with few hopeless elements, upper mouth opening wider than 50 mm, sufficient bone available for implant fixtures placement: for the edentulous maxilla the anatomical inclusion criterion was a residual ridge crest of a minimum of 4 mm wide buccolingually and higher than 10 mm high from canine to canine; for the lower maxilla a residual ridge crest at least 4 mm wide buccolingually and higher than 8 mm high in the intraforaminal area.

Exclusion criteria were: smoking and drug habits, pregnancy, irregular or thin bone crest and high smile line in the maxilla that would have needed bone reduction.

### 2.2. Clinical Procedure

#### 2.2.1. First Appointment

In the Dentistry department of the Vita-Salute University of San Raffaele, patients from both groups were examined in a preliminary oral examination.

During the appointment, after a detailed compilation of the medical and dental history, the clinicians would confirm the presence of an edentulous maxilla or treat the patients with few hopeless elements before the procedure with full mouth extractions and delivery of a temporary immediate total prosthesis.

After that, the clinician prescribed an initial orthopantomography to the patient and takes alginate impressions for the construction of occlusal rim, in order to produce a total diagnostic prosthesis correct from an aesthetic and functional point of view.

Once it was clear that a patient could be included in the clinical protocol, he or she signed a specific informed consent form for implant surgery with immediate loading.

Before the next session, the patients were divided into two groups through a randomization process: 25 patients underwent the digital protocol, and the remaining 25 was treated with the traditional protocol. Randomization processes occurred by lots in closed envelopes and were performed by a blinded operator.

#### 2.2.2. Second Appointment

The patient underwent a professional oral hygiene session of the antagonist arch. Photos of the edentulous jaw were taken (Figure 1) and the wax wall was “functionalized” using a traditional method.

#### 2.2.3. Third Appointment (Traditional Protocol)

A prosthetic device structure and functionality test and an aesthetic/phonetic evaluation test were performed. Each patient filled out a one-dimensional Verbal Rating Scale (VRS) for the assessment of his appreciation of the aesthetic test (1—very effective, 2—effective and 3—ineffective) (Figure A1). All these procedures then led to the realization of a traditional provisional prosthetic device.

#### 2.2.4. Fourth Appointment (Traditional Protocol): Surgical Phase and Immediate Loading Prosthesis

One hour before surgery the patient received 2 g of amoxicillin (Zimox, Pfizer Italia, Latina-Italy), who continued to assume 1 g twice a day for the week after surgical procedure.

After local anesthesia (4% articaine with 1:200.000 adrenaline), an incision was made starting on the center of the ridge alongside the entire length of the ridge, from the area of the first molar to the area of the first contralateral molar, with bilateral discharge incisions; a full-thickness mucoperiosteal flap was elevated and a bone remodeling were performed, if necessary to obtain a uniformly leveled bone crest.

Two-implant fixtures were inserted in the lateral alveolar areas, tilted by about 30–45 degrees relative to the occlusal plane. Then the two axial fixtures were inserted in the anterior sector (Figure 2).

Winsix TTx implants (Biosafin S.R.L., Ancona-Italy) with a diameter of 3.3 or 3.8 and length of 13 or 15 mm (Table 1) were used.

In the presence of bone with a well-represented trabecular portion, an under-preparation has been performed, to obtain a high primary stability, necessary for the following immediate loading. The insertion torque range of all implants was 35–55 N/m.

EATx Winsix extreme abutments (Biosafin S.R.L., Ancona, Italy) of 0°, 17° or 30° were screwed in at 10–20 N/m, in order to compensate for the lack of parallelism between the implants; the angle was chosen to obtain the position of the screw access hole at the occlusal or lingual level of the prosthesis. The access flap was adapted and sutured with absorbable 4–0 sutures.

At the end of the surgery, specific temporary abutments for immediate loading (EAx, Biosafin S.R.L., Ancona, Italy) were placed, mucosa was isolated with a dental dam and the prosthesis was adapted and relined directly into the patient’s mouth with cold resin.

The prosthesis was then refined and polished in the on-site laboratory, where the palatal portion was removed. Finally, the prosthetic device was screwed back in the patient’s mouth to obtain immediate loading of the implants.

#### 2.2.5. Third Appointment (Digital Protocol)

During the third visit, an occlusal rim was tested. The rim was previously functionalized according to traditional phonetic and aesthetic criteria.

The specific photographic protocol for digital planning, including intraoral and extraoral photos of the patient was performed. All the pictures were taken with the occlusal rim positioned inside the patient’s mouth with landmarks positioned on the anterior portion of the rim. These landmarks allow for the alignment of the photograph and the Standard Triangle Language (STL) ile inside the CAD Software (on both sides the canine line and the intermediate line between canine line and median line).

Two extraoral photos were also taken with a specific measurement marker positioned on the side of the patient’s face. These will be used for the realization of a two-dimensional digital project of the new smile (smile design). A VRS one-dimensional scale (1—very effective, 2—effective and 3—ineffective) for assessing the patient’s appreciation of the computerized previsualization of the prosthetic project was submitted to the patients (Figure A3).

Between the third and fourth appointments two-dimensional digital project of the new smile was realized using the Smile Lynx software (8853 S.P.A., Milan, Italy).

The scans of the edentulous model and the previously mentioned occlusal rim were obtained using a laboratory scanner (MyRay 3Di TS, Cefla, Italy). Then, the scans were matched with the 2D digital project within the CAD software (CAD Lynx 8853 S.P.A., Milan, Italy), thus allowing the three-dimensional design of the prosthesis (Figure 3). The provisional total prosthesis complete with the palatal portion was milled in PMMA (Poly(methyl methacrylate)) by a five-axis CAD/CAM milling machine (Figure 4).

#### 2.2.6. Fourth Appointment (Digital Protocol)

A mock-up test, using the provisional prosthesis, was performed trying the aesthetic appearance of the definitive prosthetic device. The patients then filled a one-dimensional VRS scale for the assessment of their appreciation of the mock-up test (Figure A1).

A specific device with the radiographic landmark (Evo-Bite with 3D-Marker, 3DIEMME, Como, Italy) was then adapted to the prosthesis directly in the oral cavity with radiotransparent silicon and delivered to the patient at the end of the appointment for the radiological exam.

Various scans were then acquired with the same spatial coordinates: one of the stereolithographic model alone, one of the temporary prothesis placed on the model and one of the prothesis on the model with the Evo bite positioned on it (3D-Marker, 3DIEMME, Como, Italy).

A CBCT (Cone Beam Computed Tomography) was prescribed to the patient. This exam had to be taken with the patient wearing the temporary prosthesis with the Evo-Bite positioned on it, including an additional radiopaque marker to be used as a reference for the following radiologic evaluation (Scan Marker, 3DIEMME, Como, Italy).

Using the RealGuide Implant Design Software (3DIEMME, Milan, Italy), the Digital Imaging and Communications in Medicine (DICOM) data of the patient’s CBCT was then matched within the STL data of the previously mentioned scans, and the virtual position of the implants was planned, based on the aesthetic prosthetic project (Figure 5 and Figure 6). The implant project was then sent to the laboratory for the realization of the stereolithographic model, which reported the exact sites for the placement of the analogs, and the surgical guide (3DIEMME, Milan, Italy) (Figure 7).

#### 2.2.7. Fifth Appointment (Digital Protocol): Surgical Phase and Immediate Loading Prosthesis

An hour before the surgery, 2 g of amoxicillin+clavulanic acid were given to the patient, which continued to assume for the following week (1 g twice a day).

After local anesthesia (4% articaine with 1:200.000 adrenaline), the surgical template was positioned and fixed in the patient’s oral cavity (Figure 8). The implants were inserted through the surgical guide, with the flapless technique, using a preordained sequence of drills dedicated to guided surgery (Figure 9). The two-implant fixtures were inserted in the lateral alveolar areas, tilted by about 30–45 degrees relative to the occlusal plane. Then the two axial fixtures were carried out in the anterior portion (Figure 1). Winsix TTx implants (Biosafin S.R.L., Ancona, Italy) with a diameter of 3.3 or 3.8, 11 or 13 mm length for the axial fixtures and 13 or 15 mm length for the tilted implants (Table 1) were used. All implants were inserted with 35–55 N/m torque.

The EATx WinSix extreme abutments (Biosafin SRL, Ancona, Italy) of 0°, 17° or 30° were screwed on at 10–20 N/cm, previously selected according to the prosthetic-implant project within the specific software for guided surgery, to offset for the lack of parallelism between implants. The angle was chosen to obtain the position of the screw access hole at the occlusal or lingual level of the prosthesis.

Specific temporary abutments (EAx, Biosafin S.R.L., Ancona, Italy) were placed, and the mucosa was isolated with a dental dam sheet. Immediate loading was then performed, positioning the provisional prosthetic device that had adapted and relined directly with pink cold resin. The device was then refined in the laboratory, where the palatal portion was removed (Figure 10). Finally, the prosthetic device was screwed back in the patient’s mouth (Figure 11).

After all the surgical-prosthetic procedures, a visual analog scale (VAS.) was submitted to both groups to evaluate pain (during and post surgery), with values from 0 (absent pain) to 10 (the maximum possible pain) (Figure A2).

#### 2.2.8. Final Prosthesis

Four months after the surgery, an impression was taken.

From the traditional group, prosthetic rehabilitations were manufactured using conventional pick-up impression. Impression transfers were screwed over the fixtures and the impression material used was Impregum (Impregum Penta, 3M Italia, Pioltello, Italy).

In the digital group, an intraoral scanner was used. Scan bodies (for TTx, Winsix, Biosafin S.R.L., Ancona, Italy) were screwed over the fixtures and splinted together. The intraoral scanner used was a Carestream CS 3500 (Version 2.5 Acquisition Software, Carestream Dental LLC, Atlanta, GA, USA).

Monolithic zirconia with vestibular ceramization final prostheses were delivered using CAD-CAM technology in both groups (Figure 12).

A final orthopantomography was prescribed to the patient (Figure 13).

### 2.3. Follow-Up

Follow-up visits were performed at 12, 24, 36 and 48 months after the surgery. These appointments provided for radiographic analysis for the evaluation of marginal bone loss. The intraoral radiographs were made with the long cone parallel technique, performing the radiography perpendicular to the longitudinal axis of the implant, using a custom occlusal model to measure the level of the marginal bone. It was then possible to measure the difference in bone level through specific software (DIGORA 2.5, Soredex, Tuusula, Finland), calibrated for each image using the implant diameter calculated on the most coronal portion of the implant neck. The linear distance between the most coronal point of the BIC. (bone–implant contact) and the coronal margin of the implant neck was measured on both mesial and distal sides, at the value closest to 0.01 mm, and then a mean value was calculated.

Besides, professional oral hygiene procedures were performed six months after implant placement and every four months after that.

### 2.4. Statistical Analysis

Dedicated software (GraphPad Prism 8.1.2, GraphPad Software Inc., California, United States) was used for statistical analysis. Peri-implant bone level measurements were reported as mean ± standard deviation values at 12, 24, 36 and 48 months. Through the one-way ANOVA test (*p* < 0.05), peri-implant bone loss was compared between the two groups at each time interval (12, 24, 36 and 48 months) and within each group by analyzing each time stage with the following ones.

## 3. Results

From March 2014 to January 2015, 50 patients were selected at the Department of Dentistry of the IRCCS Ospedale San Raffaele in Milan. A total of 200 Winsix TTx implants (Biosafin SRL, Ancona, Italy) of a diameter of 3.3 or 3.8 mm were positioned. Of them 100 were used in 25 cases of full-arch rehabilitations performed with the traditional All on Four method. The other 100 implants were used in 25 cases of full-arch rehabilitations performed with the digital method (Table 1).

All patients received a temporary prosthetic device and, after 6 months from the procedure, a definitive prosthetic device. All implants were inserted at a torque of at least 35 Ncm and were subjected to immediate loading.

### 3.1. Implant Failure and Complications

Among the patients rehabilitated according to the traditional protocol, during the first 4 months after implant insertion, 2 failures were recorded, one in the upper maxilla and one in the lower maxilla, both concerning tilted implants (Table 2). The implant fixtures were immediately replaced without compromising the prosthetic function. In patients rehabilitated with the digital protocol 100% implant survival was achieved.

A patient treated with the traditional protocol showed discomfort, pain, swelling and the presence of pus three months after surgery, while no episode of peri-implantitis, pain, paresthesia or pus was observed among the patients rehabilitated according to the digital protocol (Table 3).

Two fractures of the provisional prosthetic device were recorded for each group. Occlusal screw loosening of provisional prosthesis was observed in five cases: three were treated with a traditional method and two with the digital method. In the definitive prostheses, a 24-month and 48-month unscrewing was reported in rehabilitations performed with a traditional method, while in digitally treated patients an unscrewing at 24 months, one at 36 months and a further 48 (Table 3).

At 12 months, a case of chipping of the definitive device obtained using the traditional method was found. At 24 months a case of chipping of a definitive prosthesis obtained by the digital method was observed (Table 3). In both cases, direct repair of the existing prosthesis was performed.

### 3.2. Marginal Bone Level

The marginal bone level (MBL) was recorded during follow-up at 12, 24, 36 and 48 months (Table 4) through radiographic evaluation.

As for patients treated with the traditional protocol, the loss of peri-implant crestal bone over time has remained constant. At 48 months the mean value for bone loss for axial implants in the maxilla was 1.11 ± 0.32 mm (*n* = 30), 1.13 ± 0.24 mm for tilted implants in the maxilla (*n* = 30), 1.08 ± 0.25 for axial implants in the jaw (*n* = 20) and 1.13 ± 0.26 for jaw tilted implants (*n* = 20) (Table 4).

Bone loss in patients treated with the digital protocol at 48 months was 0.8 ± 0.10 mm for axial jaw implants (*n* = 34), 0.85 ± 10 mm for tilted jaw implants (*n* = 34), 1.08 ± 0.25 mm for axial implants in the lower maxilla (*n* = 16) and 1.13 ± 0.23 mm for tilted implants in the lower maxilla (*n* = 16) (Table 4).

The difference in the MBL between the two groups was statistically significant (*p* < 0.0001) in each time interval. The difference within each group at different time intervals was significant only between the average MBL of the digital group at 12 months compared to the same group at 36 months (*p* = 0.0066) and 48 months (*p* < 0.0001).

### 3.3. Patients’ Appreciation

Patients treated with the traditional protocol considered immediate loading with a temporary prosthesis to be very effective (95%). As for the mock-up test, 45% of the patients considered it very effective, 37% effective and 18% considered it ineffective. Traditional surgery was rated as very effective by 71% of patients and effective for the remaining 29% (Table 5). Patients treated with the digital protocol considered digital smile previsualization (93%), mock-up test (98%), guided surgery (94%) and immediate loading (92%) to be very effective (Table 5).

At the end of the surgical procedures and after seven days (Figure A4), a visual analog scale (VAS) was submitted to the patients for the evaluation of postoperative pain. All patients belonging to the group treated with the digital method, which provides flapless surgery, reported a significantly lower value of pain compared to patients treated with the traditional method.

## 4. Discussion

The aim of this study was to evaluate the survival rate of implant-prosthetic rehabilitations in patients with an edentulous arch, rehabilitated according to an entirely digital protocol, in order to understand the value of this approach in the prosthetic and surgical phases of treatment, comparing with the traditional “All on Four” method, already validated by numerous studies in the literature [5,6,7,8].

Capparé et al. and Gherlone et al. demonstrate that “All on Four” method can also be used in HIV-positive patients with a stable immune system [15,16,17].

The digital planning of the implant-prosthetic rehabilitation begins with the use of Smile Design, which allows one to obtain a two-dimensional project of the patient’s future smile. This allows a correct planning of rehabilitation in aesthetic terms, improves the interaction between specialists and communication with the patient, all of whom have been shown to appreciate the previsualization, and therefore allows a higher quality of treatment, as already described by Coachman et al. in 2017 [18].

Patients’ appreciation of digital aesthetic planning has also been described by Cattoni et al. in 2016, through the use of a VAS-type scale, which would measure the happiness of each subject with final aesthetic result of the placement of ceramic crowns and veneers in the anterior areas [19].

It has also been evaluated by Cattoni et al. in 2020 that there’s a possible neurocognitive measure of how the perception of oneself can change as a significant consequence of aesthetic prosthetic rehabilitation reduced for all the other conditions, including self-portraying pictures before the intervention, and pictures of others. Most importantly, the study reports that, among all self-retracting faces in the different stages of the prosthetic rehabilitation, those portraying the subject in her/his actual physiognomy have a somewhat special status in eliciting selectively greater brain activation in the supplementary motor area (SMA) [20].

A specific software that allows the transition from the two-dimensional previsualization of the smile to a three-dimensional volumetric study and then a CAD-CAM processing for the realization of the prosthetic product were used, as also described by Coachman et al. in 2017 [18].

It was reported in the literature in 2014 by Kapos et al. that the survival rates of crowns, abutments and superstructures made with the CAD-CAM technology are similar to those of the same manufactured with traditional methods [21].

The digital construction of the prosthetic device can be accompanied by the digital planning of the surgical procedure, due to the matching between the data of the prosthetic project and the data obtained by the CBCT, as described by several authors [22,23].

Schneider et al. in 2009 and Vinci et al. in 2020, and other authors, showed the efficacy and accuracy of computer-assisted implant surgery [24,25].

The overlap of intra- and extra-oral photographs, models, intraoral scans and CBCT is recognized as a reliable procedure by the fifth Consensus Conference of the European Association of Osseointegration of 2015 [26].

Meloni et al. in 2010 in a retrospective analysis conducted on 15 patients, described the possibility of planning implant surgery in a guided and flapless way and with immediate loading [27]. This has also been confirmed by other authors such as Komiyama et al. in 2012 [28].

The present study involves the use of a mucosal-supported surgical templates, and Gallardo et al. in 2016 and Vinci et al. in 2020 confirmed that this is a predictable procedure for implant placement [25,29].

It is widely known that a method that involves flapless implant insertion greatly reduces post-operative pain and discomfort during and after surgery, compared to open flap procedures, as also demonstrated in the present study [30,31].

Similarly, the main advantages of computer-assisted implant surgery, as already described by Hultin et al. in 2012, are the significant reduction of pain and postoperative discomfort for the patient, and the possibility of creating a temporary prosthesis to be used for the immediate functionalization of the implants [32].

The main disadvantages of guided surgery, as already described in 2009 by Schneider et al., Vinci et al. in 2020 and D’Haese et al. in 2009 are [24,25,33]:A potential damage to the bone due to insufficient irrigation;The inability to visualize the surgical anatomical landmarks;The increased risk of error in implant positioning with increasing degrees of maxillary bone atrophy;A disparity between the virtual plan and the actual position of the implant in the oral cavity at the end of the surgery;Difficulty in positioning the surgical template both during the CBCT Scan and during the surgical procedures.

As Malo et al. said in 2007, there are several contraindications, which include: insufficient bone volume, remaining teeth that interfere with the planning for implant placement, insufficient mouth opening to accommodate surgical instrumentation of at least 50 mm or bone reduction needed due to a high smile line in the maxilla, irregular bone crest or thin bone crest [34]. Inclusion criteria of the present study included all these contraindications, especially insufficient bone volume.

Accuracy and predictability of the intraoral scanner for implant full-arch rehabilitations are demonstrated by many authors so digital impressions is a viable alternative to analog techniques [35,36].

The levels of peri-implant bone loss obtained in the present study have proved to be similar to those reported by other authors in the literature, both for the group of patients treated with traditional surgery, and for the group treated with guided surgery [7,34,37].

The present clinical trial has some limitations, the main one is the follow-up. This type of studies would need longer follow-up. Further studies with a larger number of patients are also needed.

## 5. Conclusions

The obtained results show that the present protocol that is entirely digital, represents a valid therapeutic alternative to the traditional “All on Four” protocol for implant-supported rehabilitation of edentulous arches. However more long-term prospective clinical trials are needed to confirm the effectiveness of the surgical-prosthetic protocols used in this study and it is good not to underestimate the design difficulties: to be successful, a broad knowledge and mastery of topographical anatomy, radiographic imaging, surgical techniques and prosthetic procedures are essential.

It is necessary to select more carefully the clinical cases subject to both methods, as described previously in the inclusion and exclusion criteria.

Ultimately with the evolution of technologies, it is hoped that a digital workflow can be further simplified and increasingly within reach of each clinician.

## Figures and Tables

**Figure 1 ijerph-18-03449-f001:**
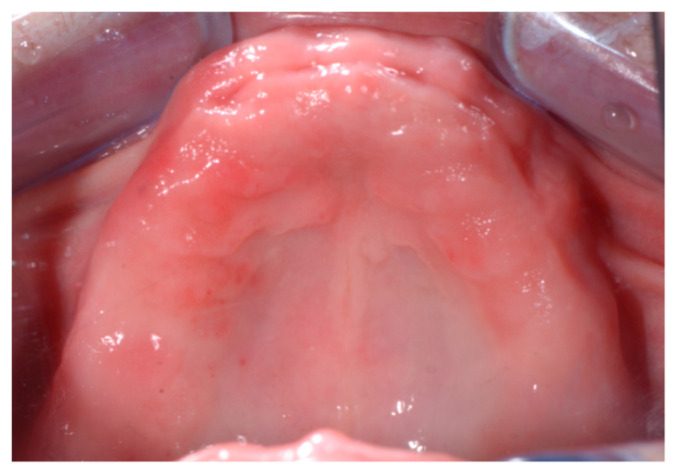
Edentulous maxilla.

**Figure 2 ijerph-18-03449-f002:**
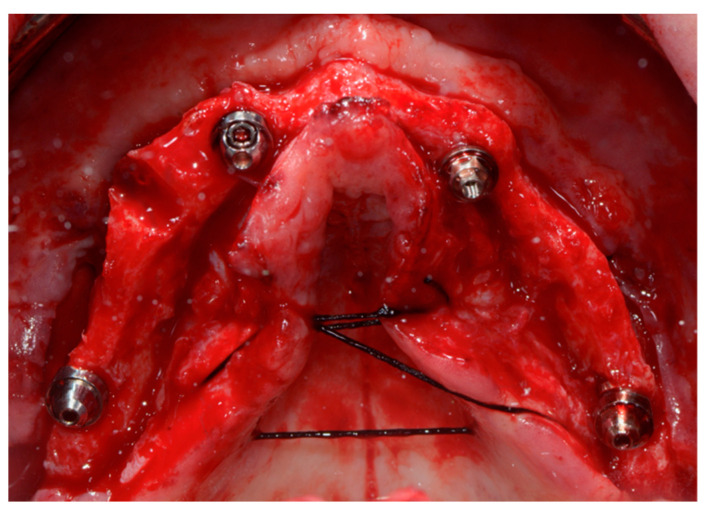
Open flap surgery for implant positioning (traditional protocol).

**Figure 3 ijerph-18-03449-f003:**
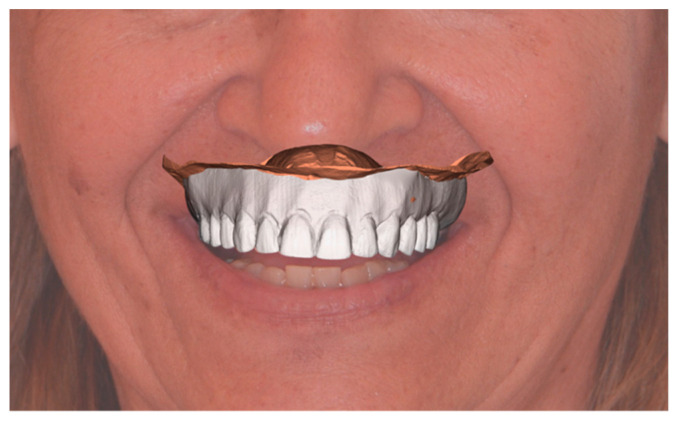
Three-dimensional digital project.

**Figure 4 ijerph-18-03449-f004:**
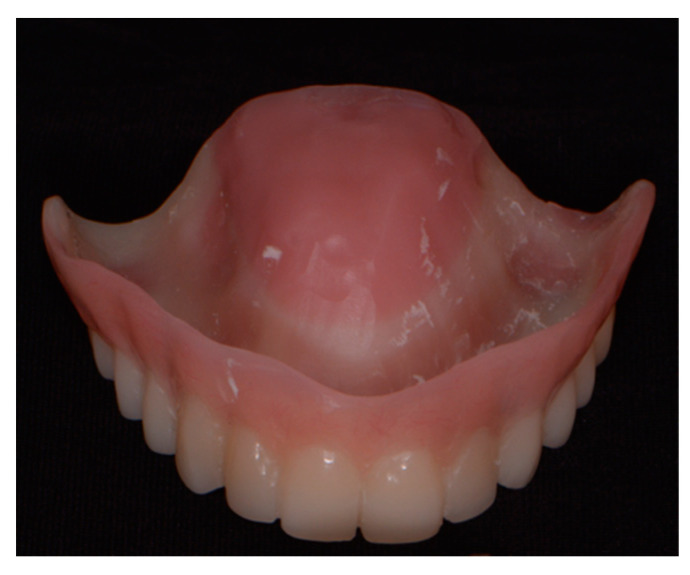
PMMA total provisional prosthetic device.

**Figure 5 ijerph-18-03449-f005:**
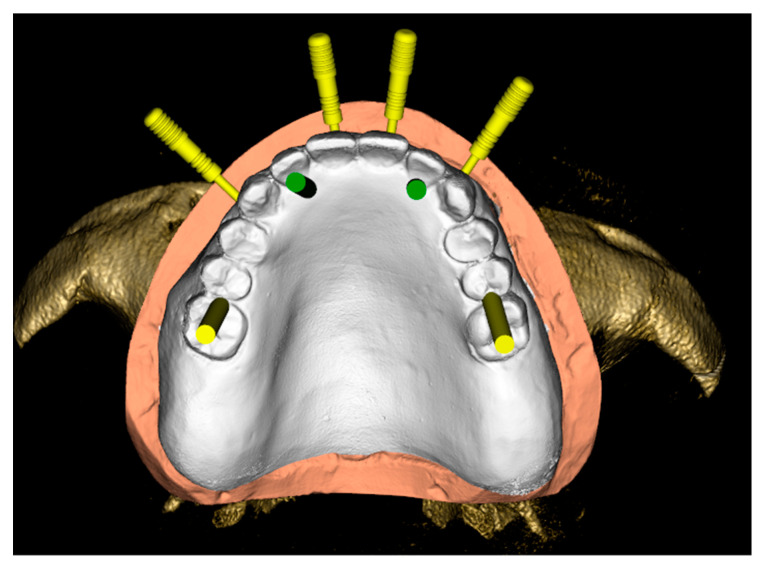
Virtual positioning of the implants, based on the aesthetic prosthetic project.

**Figure 6 ijerph-18-03449-f006:**
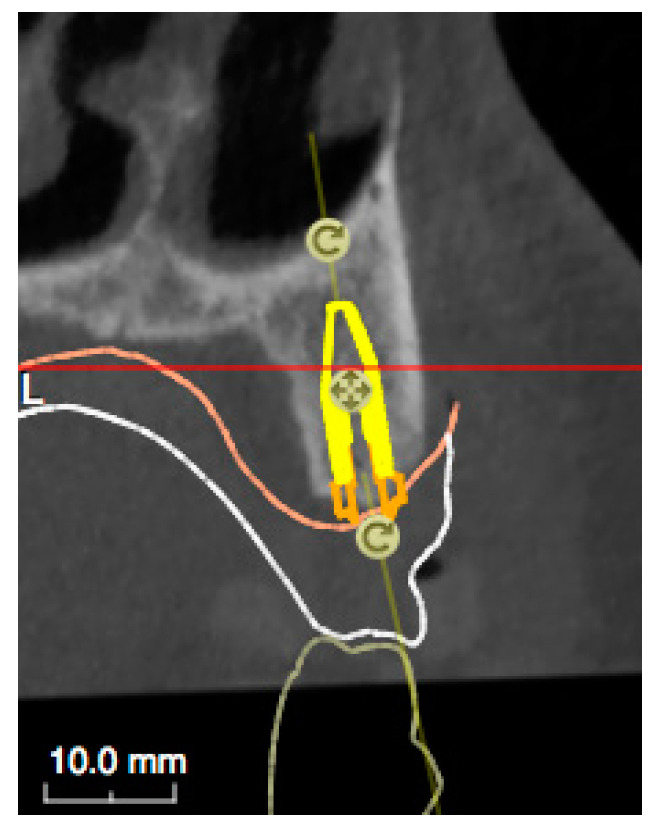
Virtual positioning of the implants, based on the aesthetic prosthetic project.

**Figure 7 ijerph-18-03449-f007:**
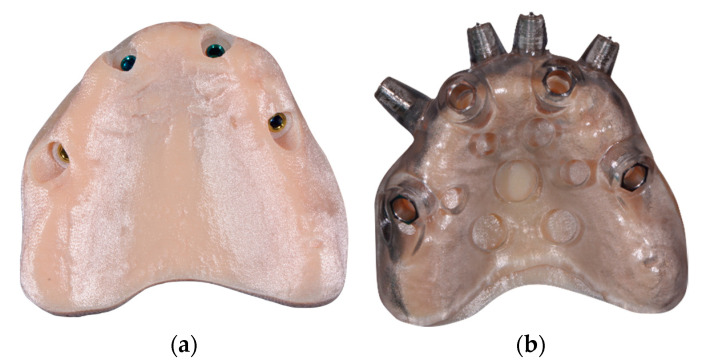
(**a**) Stereolithographic model with analogs. (**b**) Surgical guide.

**Figure 8 ijerph-18-03449-f008:**
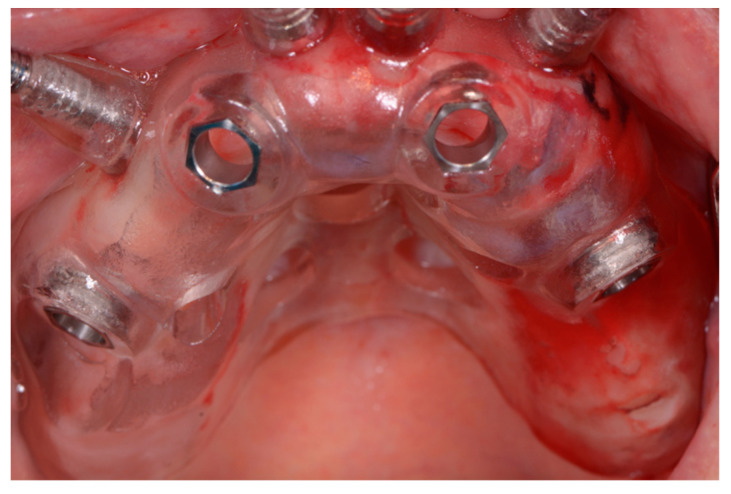
Surgical template positioned and fixed in the oral cavity.

**Figure 9 ijerph-18-03449-f009:**
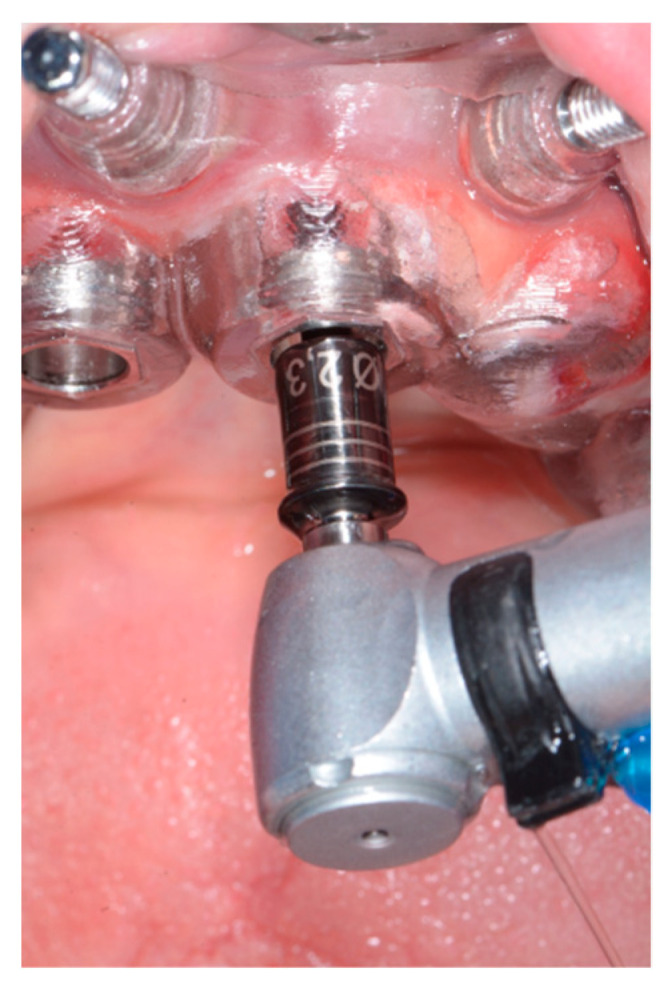
Flapless surgery.

**Figure 10 ijerph-18-03449-f010:**
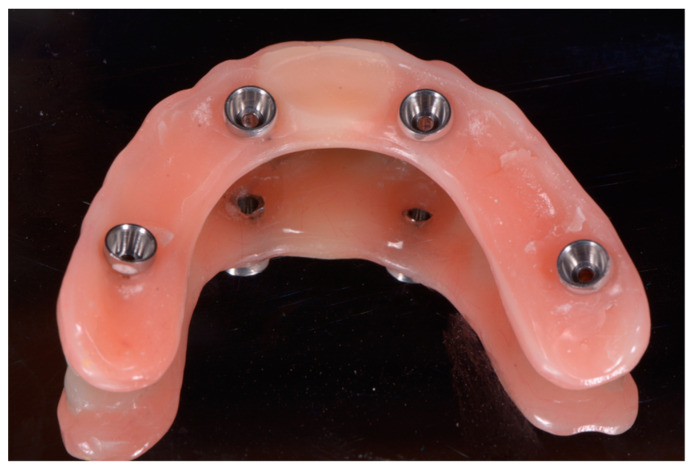
Adapted, relined and refined provisional prosthesis.

**Figure 11 ijerph-18-03449-f011:**
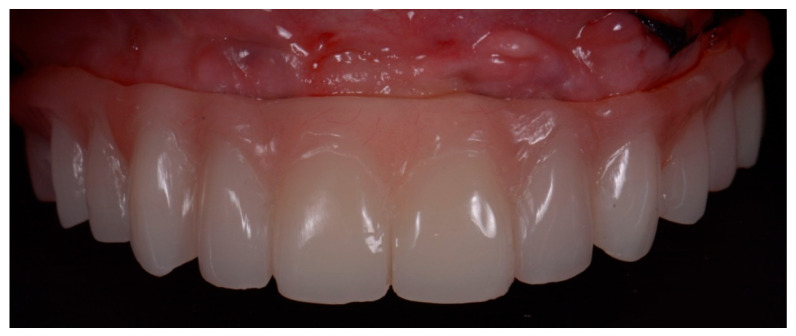
Provisional prosthesis screwed in the patient’s mouth.

**Figure 12 ijerph-18-03449-f012:**
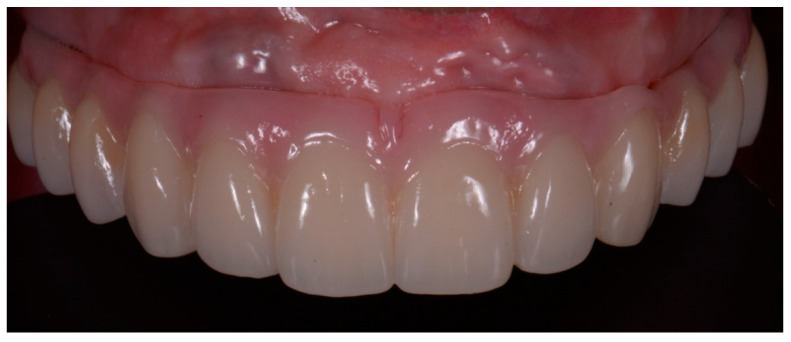
Monolithic zirconia final prosthesis.

**Figure 13 ijerph-18-03449-f013:**
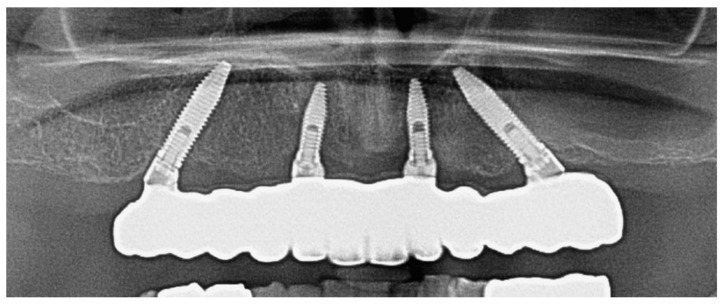
Final orthopantomography.

**Table 1 ijerph-18-03449-t001:** Implants’ diameters (D) and lengths (L).

			L 11 mm	L 13 mm	L 15 mm
Traditional protocol (*n* = 100)	Upper maxilla (*n* = 60)	D 3.3	0	16	0
		D 3.8	0	27	17
	Lower maxilla (*n* = 40)	D 3.3	0	12	0
		D 3.8	0	19	9
Digital protocol (*n* = 100)	Upper maxilla (*n* = 68)	D 3.3	6	14	0
		D 3.8	12	32	6
	Lower maxilla (*n* = 32)	D 3.3	0	6	0
		D 3.8	4	22	0

**Table 2 ijerph-18-03449-t002:** Survival rate.

		Implants Positioned	Failed Implants	Implant Survival (%)
Traditional Protocol				
Upper maxilla (*n* = 60)	Axial	30	0	100%
	Tilted	30	1	96.67%
Lower maxilla (*n* = 40)	Axial	20	0	100%
	Tilted	20	1	95.00%
Digital protocol				
Upper maxilla (*n* = 68)	Axial	34	0	100%
	Tilted	34	0	100%
Lower maxilla (*n* = 32)	Axial	16	0	100%
	Tilted	16	0	100%

**Table 3 ijerph-18-03449-t003:** Implant failure and complications.

	Traditional Protocol								Digital Protocol							
Months	12		24		36		48		12		24		36		48	
	*n*	%	*n*	%	*n*	%	*n*	%	*n*	%	*n*	%	*n*	%	*n*	*%*
Implant failures	2	2%	0	0	0	0	0	0	0	0	0	0	0	0	0	0
Peri-implantitis	1	1%	0	0	0	0	0	0	0	0	0	0	0	0	0	0
Fixture fractures	0	0	0	0	0	0	0	0	0	0	0	0	0	0	0	0
Unscrewing	3	3%	1	1%	0	0	1	1%	2	2%	1	1%	1	1%	1	1%
Provisional prosthetic fractures	2	n.a	/		/		/		2	n.a	/		/		/	
Definitive prosthetic Chipping	1	n.a	0	0	0	0	0	0	0	0	1	n.a	0	0	0	0
Pus	1	n.a	0	0	0	0	0	0	0	0	0	0	0	0	0	0
Pain	1	n.a	0	0	0	0	0	0	0	0	0	0	0	0	0	0
Paresthesia	0	0	0	0	0	0	0	0	0	0	0	0	0	0	0	0

**Table 4 ijerph-18-03449-t004:** Marginal bone level.

		12 Months	24 Months	36 Months	48 Months
Traditional protocol					
		mm	mm	mm	mm
Upper maxilla	Axial (*n* = 30)	1.02 ± 0.33	1.08 ± 0.34	1.10 ± 0.32	1.11 ± 0.32
	Tilted (*n* = 30)	1.05 ± 0.27	1.08 ± 0.26	1.11 ± 0.25	1.13 ± 0.24
Lower maxilla	Axial (*n* = 20)	1.04 ± 0.28	1.05 ± 0.26	1.06 ± 0.26	1.08 ± 0.25
	Tilted (*n* = 20)	1.05 ± 0.29	1.09 ± 0.25	1.12 ± 0.23	1.13 ± 0.23
Total	*n* = 100	1.04 ± 0.29	1.08 ± 0.28	1.10 ± 0.27	1.12 ± 0.26
Digital protocol					
Upper maxilla	Axial (*n* = 34)	0.65 ± 0.10	0.72 ± 0.13	0.76 ± 0.11	0.8 ± 0.10
	Tilted (*n* = 34)	0.69 ± 0.11	0.78 ± 0.11	0.81 ± 0.11	0.85 ± 0.10
Lower maxilla	Axial (*n* = 16)	0.69 ± 0.19	0.73 ± 0.16	0.79 ± 0.14	0.82 ± 0.15
	Tilted (*n* = 16)	0.71 ± 0.14	0.77 ± 0.11	0.80 ± 0.10	0.84 ± 0.10
Total	*n* = 100	0.68 ± 0.13	0.75 ± 0.13	0.79 ± 0.11 *	0.83 ± 0.11 *
*p* value *(Total Traditional* vs. *Total Digital)*		<0.0001	<0.0001	<0.0001	<0.0001

* *p* < 0.05 vs. 12 months digital group.

**Table 5 ijerph-18-03449-t005:** Patients’ appreciation.

	Very Effective	Effective	Ineffective
Traditional protocol			
Mock-up test	45%	37%	18%
Traditional surgery	71%	29%	0%
Immediate loading	95%	5%	0%
Digital protocol			
Digital smile previsualization	93%	7%	0%
Mock-up test	98%	2%	0%
Guided surgery	94%	6%	0%
Immediate loading	92%	8%	0%

## Data Availability

Not applicable.

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
