# Peer review of "Digital Smile Designed Computer-Aided Surgery versus Traditional Workflow in “All on Four” Rehabilitations: A Randomized Clinical Trial with 4-Years Follow-Up"

_ijerph, 2021, doi:10.3390/ijerph18073449_

Round 1

Reviewer 1 Report

This is a well written interesting paper. this paper compares the traditional all on four rehabilitation to a fully digital cad cam assisted and computer smile design all on 4 techniques.

the datas are well organized and well presented. the main strength is randomization in two groups which appear homogenous. the paper is well written with nice photos.

the main weakness are:

 1. follow up

2. only a radiological parameter is used to describe implant success

3 .there is no evaluation of costs  needed for the two methods

4. the methods used for evaluation , normally should be described in the methods session , while in the results session the authors should detail only the results.

Therefore, I think the authors should tell how many appointments are needed for each method. i think they should also precise the impact on costs for the patients for both methods.

The author should organize better the material and methods session. in the result session only the result of the evaluation should be given 3 the authors should say something about the cist of the two methods

The paper could be accepted also in the present form , but keeping in mind these recommendations.

Author Response

Thank you for the recommendations.

  1. We know that follow up is a weakness of this paper. In fact we keep following the patients for an eventual future paper.
  2. Implant success has been described by highlighting implant complications.
    And we preferred to use MBL through radiographical evaluation as a parameter, because in our opinion it is the most repeatable one.
  3. We did not considered the costs in this paper. Certainly there are higher costs for the doctor and for the patient with the digital protocol.
  4. Line 321. I canceled the sentence. It was actually a repetition.
  5. Numbers of appointments: materials and methods is divided by appointments.

Reviewer 2 Report

The present study has an original purpose and an interesting clinical value.

Its results may be of interest both to readers who deal with scientific research and to clinicians.

Only minor revisions of English are required by mother tongue.

INTRODUCTION:

Clear and well written.

The state of the problem is addressed.

The objectives of the study and the current state of the literature on the subject are defined.

M & M / RESULTS:

Clear and detailed.

Also, the manuscript includes clear and detailed tables and high-quality images.

DISCUSSION:

This section is well written and comprehensive, with adequate explanations of the results obtained and clear comparisons with the results that are already present in the literature.

It should be clarified that Further experimental studies with a larger number of patients should be made to confirm the outcomes of the present investigation.

Author Response

Thank you for your response.

I have added Line 418, as you suggested.

Reviewer 3 Report

This is a clinical research article that compared different therapeutic methods of 25 individuals in each group. The originality and significance of the content are high. However, the content is not relevant between environmental quality and public health.

Author Response

Thank you for your response.

I know that, but it was suggested to us. However it is in the Oral Health section.

Reviewer 4 Report

The authors are to be thanked for their considerable efforts in performing this extensive study. 
There are a number of typos and grammatical errors for which someone experienced in reviewing written English manuscripts is sufficiently proficient at catching these minor errors.
The number of tables should be reduced and summarized.  Table 1, 2, and 3.  Table 3  could be reported on the text.
For analysis, authors could perform an ANOVA for repeated measurements, instead of One-Way ANOVA, in order to control for the marginal bone level for individual change between follow-up measurements.  Tables must include de p values or 95% confidence intervals and the statistical test used at the bottom of the table.

Consort guidelines are strongly suggested for authors. The inclusion of The CONSORT Flow Diagram will enhance the clinical trial design.

Author Response

Thank you for your response.

  1. I did not understand “tables should be reduced and summarized”.
  2. Given the small number of patients, we thought One-Way ANOVA was more convenient.
  3. Table 4 is now correct with p values.
  4. Consort flow diagram added to the manuscript [line 515]

Reviewer 5 Report

Dear Authors,

After reading your manuscript I kindly recommend the following modifications:

1- Bibliography is not correctly organized. You are missing Roman and numerical system in the same text. Please correct 

2- English must be improved as you have some examples in lines 16,93,94,290.

3- What kind of PMMA material and color was used to mill the provisional prosthesis? Was it used acrylic for distinguish teeth and gingiva?

4- Was it necessary during the surgical procedure to apply vertical osteotomy to avoid the exposure of denture-natural gingiva of the patient? This is a common step described in the original All-on-4 protocol. If so how did you manage it in the digital group?

5- How did you calibrate the intra-oral X-rays in follow-up session? Can you describe with more detail the protocol used? Is it possible to see some example?

6- It is important to introduce your results in the discussion and compare it with other authors with similar protocols, concerning bone loss measurements.

Thank you very much in advance.

Regards

Author Response

Thank you for your response.

  1. Bibliography is now correct.
  2. Minor corrections done.
  3. The provisional prothesis was milled in PMMA (Temp Premium, Zirkonzahn, Gais, Italy) in a single color. The gingiva was then manually addicted and teeth were manually painted to fit the patient’s color by the technician.
  4. This this is one of the disadvantages of flapless surgery, as well as irregular bone crest. For this reason one of the exclusion criteria is “irregular or thin bone crest and high smile line in the maxilla that would have needed bone reduction” [line 84-85].
  5. The protocol is well described in lines 259-267. We used a custom occlusal model made in silicon material (Elite HD Putty Soft, Zhermack).
  6. You can find it in lines 418-420.

Round 2

Reviewer 5 Report

Dear Authors,

1- It is still unclear how did you fabricate the custom device for the intraoral X-rays and the bone loss measurements. Being the most important objective proposed in this research it is not acceptable that the description of the methodology is so short.

I would like to see x-ray examples, the device and how it was built, and measurement examples.

2- The comparison in the discussion with other authors is still very short.

Best regards.

Author Response

Dear reviewer,
I am sorry for the late reply.
I am sending you an x-ray example and the custom device used. It is made with Rinn centrator and silicon based putty.

This manuscript is a resubmission of an earlier submission. The following is a list of the peer review reports and author responses from that submission.

Round 1

Reviewer 1 Report

First of all, I want to applaud the efforts of the authors who have recruited many patients for good research and followed-up for a long time.

However, there are various problems in this study, and there are shortcomings that are not suitable for publication.

First of all, this manuscript does not fit the format of MDPI journal at all. The basic format suggested by the journal for publication was never followed. The abstract is too long, and the location of the reference and the format of the reference have not been preserved.

In addition, the exact protocol of this study was not presented. It has not been clearly suggested what differences exist between the traditional and digital methods. It was not suggested how the process of taking impressions for prosthesis fabrication proceeded. If the digital method is computer-based planning for guided implant surgery in advance and applying smile design, that part should be reflected in the title.

Guided implant surgery was used in the digital group of this study. In the case of completely edentulous patients, the accuracy of the soft tissue supported surgical guide is known to be very low. Was the implant accurately placed in the desired location during the surgical procedure? Did the inaccuracy of the guide not affect the results?

Discussion is too weak, and the results of this study have not been sufficiently considered. Of the numerous all-on-four studies published so far, there are too many studies to compare with this one. And it is difficult to understand what this study means to readers and what clinical implications are there.

Author Response

First of all, I want to applaud the efforts of the authors who have recruited many patients for good research and followed-up for a long time.

However, there are various problems in this study, and there are shortcomings that are not suitable for publication.

First of all, this manuscript does not fit the format of MDPI journal at all. The basic format suggested by the journal for publication was never followed. The abstract is too long, and the location of the reference and the format of the reference have not been preserved.

Format of the journal done. I also modified the references. And I reduced the abstract, now it is shorter.

In addition, the exact protocol of this study was not presented. It has not been clearly suggested what differences exist between the traditional and digital methods. It was not suggested how the process of taking impressions for prosthesis fabrication proceeded.

Thank you, we forgot to describe the impressions for the final prosthesis.

If the digital method is computer-based planning for guided implant surgery in advance and applying smile design, that part should be reflected in the title.

We have decided to modify the title, as you advised: “Digital Smile Designed Computer-Aided Surgery Versus Traditional Workflow in “All on Four” Rehabilitations: A Randomized Clinical Trial with 4-years Follow-Up”.

Guided implant surgery was used in the digital group of this study. In the case of completely edentulous patients, the accuracy of the soft tissue supported surgical guide is known to be very low. Was the implant accurately placed in the desired location during the surgical procedure? Did the inaccuracy of the guide not affect the results?

No, in our opinion the soft tissue supported surgical guide is not inaccurate. I also add a reference in the discussion about it: https://www.ncbi.nlm.nih.gov/pmc/articles/PMC7141387/
“Vinci, R.; Manacorda, M.; Abundo, R.; Lucchina, A.; Scarano, A.; Crocetta, C.; Lo Muzio, L.; Gherlone, E.; Mastrangelo, F. Accuracy of Edentulous Computer-Aided Implant Surgery as Compared to Virtual Planning: A Retrospective Multicenter Study. J. Clin. Med. 2020, 9, 774.”

Discussion is too weak, and the results of this study have not been sufficiently considered. Of the numerous all-on-four studies published so far, there are too many studies to compare with this one. And it is difficult to understand what this study means to readers and what clinical implications are there.

There are not many studies that compare the traditional All on four with a total digital workflow. Our goal is to reach more and more clinicians and explain to them that this is a valid alternative. Of course with some difficulties.

Reviewer 2 Report

Text format: Are you sure that the references should be reported at the end of each page? Please check IJERPH guidelines. References are not formatted in the main manuscript properly. Please check IJERPH guidelines
Authors: After Luca Chirico there is a dot. Please read the article carefully and check the presence of typos. I would suggest a professional language editing service.
Introduction:
page 2, lines 47-54: I would suggest to remove these sentences.
page 2, line 53: there should be a dot instead of ";"
page 2, line 57: "four" instead of "4"
page 3, line 70: please add a reference for this sentence (as a suggestion, please find an article below)
How to obtain an orthodontic virtual patient through superimposition of three-dimensional data: A systematic review
page 3, line 72: please remove the coma between subject and verb
page 3, line 72: "More" without capital letter
Materials and Method: Were the patients consecutively enrolled? Did you calculate the sample size? How patients' allocation was randomized?
page 3, line 92: "buccolingually" instead of "buccolingual"
page 3, line 92: "higher than 10 mm" instead of "greater than 10 mm high". Do the same for line 93.
page 3, line 97: "uncompensated" instead of "decompensated"
page 3, line 98: I would remove this sentence.
page 4, line 106: "prescribed" instead of "would prescribe"
page 4, line 107: "Alginate" without capital letter
page 4, line 110: Informed consent without capital letter.
Please provide the ethical board approval ID number.
page 4, line 131: please provide a reference for antibiotic prophylaxis with all on four surgery.
page 4, line 134: "a full-thickness mucoperiosteal flap was elevated" instead of "a full-thickness mucoperiosteal flap with exposure the bone of the edentulous maxilla"
page 4, lines 135-136: please rephrase this sentence.
page 4, line 137: "inserted" instead of "carry on"
page 4, line 139: "used." instead of "used;"
page 4, line 140: "In presence" instead of "In the presence"

Further Suggestions

Title and Abstract: I would suggest to be more precise about the aim of the study, such as "Radiographic outcome, self-perceived satisfaction, complication/failure rate of Digital Versus Traditional Workflow in “All on 2 Four” Rehabilitations: A Randomized Clinical Trial 3 with 4-years Follow-Up

Terms:  "All on four" is a trademark, I think you should use "full arch" rehabilitation. You can use "implant-supported" (page 1, line 64) instead of "implant supported" rehabilitation ((page 1, line 47). Try to use the same terms for the whole manuscript.

Keywords: the keywords are missing  

Introduction: I would try to rephrase the introduction, focusing on the title topic. You could add some information regarding the radiographical success of in “All on  Four” Rehabilitations, or self-perceived satisfaction, and the type and rate of complications/failures. You can move some sentences from the discussion to the introduction about the advantages of traditional (if available) and digital workflow for a prosthetic rehabilitation. You should specify that "digital protocol" involves a flapless guided-surgery and the realization of prostheses, while the "traditional approach" involves open flap surgery and conventional realization/construction of prostheses.

Methods: The self-perceived satisfaction has been assessed before the delivery of a traditional provisional denture? The same for the digital process? Can you motivate this choice? In this sense, it seems that the patients can evaluate just the procedure, not the fitting of the final prosthesis. You can add in the discussion that this could represent a limitation for the assessment of traditional/digital approach. pg.4, line 138: you can remove "only" Please, provide the questionnaire as supplementary material. Please specify the abbreviation - "VRS" assessment scale. It it important to cite the extremes of the scale, whether the questionnaire was performed online, and the range (if VRS is a numerical rating scale); as a suggestion, you can cite the article below:   Staderini E, De Luca M, Candida E, Rizzo MI, Rajabtork Zadeh O, Bucci D, Zama M, Lajolo C, Cordaro M, Gallenzi P. Lay People Esthetic Evaluation of Primary Surgical Repair on Three-Dimensional Images of Cleft Lip and Palate Patients. Medicina (Kaunas). 2019 Sep 8;55(9):576. doi: 10.3390/medicina55090576. PMID: 31500380; PMCID: PMC6780772.   page 8, lines 232-233: the pain was assessed with the same scale of patients satisfaction? Please provide some details. The complication and failure rate is not reported anywhere; you can specify the definition of "failure" and "complication" and the classification (prosthetic failure/implant failure).

Text format: The statistical analysis has a different text format

Results: Page 13, lines 311-313: this sentence should be moved to Materials and Methods.

Discussion: Page 14, line 347: "Intra-" instead of "Intra". I would suggest to be more focused on the interpretation of the results.

Conclusion: page 16, lines 388-390: this sentence should be rephrased.

Author Response

  • Were the patients consecutively enrolled?

The patients were randomly enrolled when they came in our department asking a full fixed rehabilitation.

  • Did you calculate the sample size?

We enrolled as many patients as possible for this study, in order to obtain a significative clinical result.

  • Page 3, line 94: “irregular or thin bone crest and high smile line in the maxilla that would have needed bone reduction”

We do not remove this sentence because, in our opinion, it is an important exclusion criteria.

Text format: Are you sure that the references should be reported at the end of each page? Please check IJERPH guidelines. References are not formatted in the main manuscript properly. Please check IJERPH guidelines.
I modified position and format of the references.
Authors: After Luca Chirico there is a dot. Please read the article carefully and check the presence of typos. I would suggest a professional language editing service.
Introduction:
page 2, lines 47-54: I would suggest to remove these sentences. done
page 2, line 53: there should be a dot instead of ";" done
page 2, line 57: "four" instead of "4" done
page 3, line 70: please add a reference for this sentence (as a suggestion, please find an article below) done, thank you
How to obtain an orthodontic virtual patient through superimposition of three-dimensional data: A systematic review
page 3, line 72: please remove the coma between subject and verb done
page 3, line 72: "More" without capital letter done
Materials and Method: Were the patients consecutively enrolled? Did you calculate the sample size? How patients' allocation was randomized?

The patients were randomly enrolled when they came in our department asking a full fixed rehabilitation. We enrolled as many patients as possible for this study, in order to obtain a significative clinical result.
page 3, line 92: "buccolingually" instead of "buccolingual" done
page 3, line 92: "higher than 10 mm" instead of "greater than 10 mm high". Do the same for line 93. done
page 3, line 97: "uncompensated" instead of "decompensated" done
page 3, line 98: I would remove this sentence. “irregular or thin bone crest and high smile line in the maxilla that would have needed bone reduction” We do not remove this sentence because, in our opinion, it is an important exclusion criteria.
page 4, line 106: "prescribed" instead of "would prescribe" done
page 4, line 107: "Alginate" without capital letter done
page 4, line 110: Informed consent without capital letter. done
Please provide the ethical board approval ID number.

Ethical board approval number CE/INT/10/2015
page 4, line 131: please provide a reference for antibiotic prophylaxis with all on four surgery.

I did not find bibliography about antibiotic prophylaxis in all on four surgery.
page 4, line 134: "a full-thickness mucoperiosteal flap was elevated" instead of "a full-thickness mucoperiosteal flap with exposure the bone of the edentulous maxilla" done
page 4, lines 135-136: please rephrase this sentence. done
page 4, line 137: "inserted" instead of "carry on" done
page 4, line 139: "used." instead of "used;" done
page 4, line 140: "In presence" instead of "In the presence" done

Further Suggestions

Title and Abstract: I would suggest to be more precise about the aim of the study, such as "Radiographic outcome, self-perceived satisfaction, complication/failure rate of Digital Versus Traditional Workflow in “All on Four” Rehabilitations: A Randomized Clinical Trial 3 with 4-years Follow-Up

I have already changed the title, as another reviewer suggested me. Now the new title is:

“Digital Smile Designed Computer-Aided Surgery Versus Traditional Workflow in “All on Four” Rehabilitations: A Randomized Clinical Trial with 4-years Follow-Up “

Terms:  "All on four" is a trademark, I think you should use "full arch" rehabilitation.

We used the all on four technique, so I think it is correct to use “All on Four”.

You can use "implant-supported" (page 1, line 64) instead of "implant supported" rehabilitation ((page 1, line 47). Try to use the same terms for the whole manuscript. Done. I am using “implant-supported”

Keywords: the keywords are missing. Added.

Introduction: I would try to rephrase the introduction, focusing on the title topic. You could add some information regarding the radiographical success of in “All on  Four” Rehabilitations, or self-perceived satisfaction, and the type and rate of complications/failures. You can move some sentences from the discussion to the introduction about the advantages of traditional (if available) and digital workflow for a prosthetic rehabilitation. You should specify that "digital protocol" involves a flapless guided-surgery and the realization of prostheses, while the "traditional approach" involves open flap surgery and conventional realization/construction of prostheses.

I have added this part (page 2, line 58-59).

Methods: The self-perceived satisfaction has been assessed before the delivery of a traditional provisional denture? The same for the digital process? Can you motivate this choice? In this sense, it seems that the patients can evaluate just the procedure, not the fitting of the final prosthesis. You can add in the discussion that this could represent a limitation for the assessment of traditional/digital approach. We did not evaluate the patients’ appreciations of the provisional and final prosthesis.

pg.4, line 138: you can remove "only" done

Please, provide the questionnaire as supplementary material. Please specify the abbreviation - "VRS" assessment scale. It it important to cite the extremes of the scale, whether the questionnaire was performed online, and the range (if VRS is a numerical rating scale); as a suggestion, you can cite the article below:   Staderini E, De Luca M, Candida E, Rizzo MI, Rajabtork Zadeh O, Bucci D, Zama M, Lajolo C, Cordaro M, Gallenzi P. Lay People Esthetic Evaluation of Primary Surgical Repair on Three-Dimensional Images of Cleft Lip and Palate Patients. Medicina (Kaunas). 2019 Sep 8;55(9):576. doi: 10.3390/medicina55090576. PMID: 31500380; PMCID: PMC6780772.

I added the extremes of the VRS scale on page 11, line 306.

I am not able to find the questionnaire right now, I do not have the computer with all the documents with me.

 page 8, lines 232-233: the pain was assessed with the same scale of patients satisfaction? Please provide some details.

It was already explained, but in the results. Now it is in Materials and Methods (pag 7, line 221-223)

The complication and failure rate is not reported anywhere; you can specify the definition of "failure" and "complication" and the classification (prosthetic failure/implant failure).

All the failures are described in table 3.

Text format: The statistical analysis has a different text format.

Results: Page 13, lines 311-313: this sentence should be moved to Materials and Methods. done

Discussion: Page 14, line 347: "Intra-" instead of "Intra".  done I would suggest to be more focused on the interpretation of the results.

Conclusion: page 16, lines 388-390: this sentence should be rephrased. done

Reviewer 3 Report

This is a worthwhile paper and should be published more or less as submitted. However, there are some minor editorial changes that must be made before final approval can be given.

In the Abstract, there is no consistency on the way "All On Four" is written.  Should every word start with a capital letter (They don't always), should there be quotation marks (not always present), should the middle word be "On" (lines 11, 12, 23 and 43) or "of" (line 31)?  Please look at this again, and make it consistent.

Line 53: Replace ; with a full stop.

Line 70. Do something about the spacing. Also, lines 136, 149, 213, 233, 253.

Line 206: Patients should be "he or she" (or possibly "they" as this is gender-neutral).

Lines 257-262 need to be in a smaller font to match the rest of the text.

Line 339, last word "in" should begin with a lower-case letter.

Lines 363-370: There should be consistency in use of capital letters at the beginning of sentences.

Author Response

Comments and Suggestions for Authors

This is a worthwhile paper and should be published more or less as submitted. Thank you! However, there are some minor editorial changes that must be made before final approval can be given.

In the Abstract, there is no consistency on the way "All On Four" is written.  Should every word start with a capital letter (They don't always), should there be quotation marks (not always present), should the middle word be "On" (lines 11, 12, 23 and 43) or "of" (line 31)?  Please look at this again, and make it consistent. Done.

Line 53: Replace ; with a full stop. Done

Line 70. Do something about the spacing. Also, lines 136, 149, 213, 233, 253. Done

Line 206: Patients should be "he or she" (or possibly "they" as this is gender-neutral). Done

Lines 257-262 need to be in a smaller font to match the rest of the text. Done

Line 339, last word "in" should begin with a lower-case letter. Done

Lines 363-370: There should be consistency in use of capital letters at the beginning of sentences. Done

Reviewer 4 Report

The subject of the article is very interesting and related to the modern implantology. The article was written in a correct and comprehensive language, the English is understandable, and the results provide an advance in current knowledge. The data and analyses are presented appropriately. In my opinion the highest standards for presentation of the results are used and the conclusions are interesting for the readership of the Journal. The results of the conducted tests were presented correctly and clearly. Numerous analyzes of the assessed parameters were used in the research. The literature review is rich and closely related to the subject of the article. The tables in the article help to understand the complexity of research.

Nevertheless some changes are necessary.

The inclusion and exclusion criteria for patients are written incorrectly.

The exclusion criterion applies only to patients who meet the inclusion criterion, but for some important and extraordinary reason cannot take part in the studies. So, information, such as ; good general health, without chronic diseases ( like immunosuppression, untreated coagulation 95 problems, chemotherapy and radiotherapy, assumption of bisphosphonate drugs, cardiac conditions), - should be in a group of inclusion criteria. The exclusion criteria can contain; smoking,  drug habit, pregnancy, irregular or thin bone crest and high smile line in the maxilla that would have needed bone reduction.

Please collect the references at the end of manuscript, not at the end of each paper

please change;

lateral alveolar areas (not posterior)

prosthetic device - denture

lower maxilla - mandible

please specify which method is used to randomly allocate patients (e.g. Excell or draw)

please standardize the font size

Author Response

Comments and Suggestions for Authors

The subject of the article is very interesting and related to the modern implantology. The article was written in a correct and comprehensive language, the English is understandable, and the results provide an advance in current knowledge. The data and analyses are presented appropriately. In my opinion the highest standards for presentation of the results are used and the conclusions are interesting for the readership of the Journal. The results of the conducted tests were presented correctly and clearly. Numerous analyzes of the assessed parameters were used in the research. The literature review is rich and closely related to the subject of the article. The tables in the article help to understand the complexity of research. Nevertheless some changes are necessary. Thank you!

The inclusion and exclusion criteria for patients are written incorrectly. The exclusion criterion applies only to patients who meet the inclusion criterion, but for some important and extraordinary reason cannot take part in the studies. So, information, such as ; good general health, without chronic diseases ( like immunosuppression, untreated coagulation 95 problems, chemotherapy and radiotherapy, assumption of bisphosphonate drugs, cardiac conditions), - should be in a group of inclusion criteria. The exclusion criteria can contain; smoking,  drug habit, pregnancy, irregular or thin bone crest and high smile line in the maxilla that would have needed bone reduction. Done

Please collect the references at the end of manuscript, not at the end of each paper. Done

please change;

lateral alveolar areas (not posterior) Done

prosthetic device – denture Done

lower maxilla – mandible Done

please specify which method is used to randomly allocate patients (e.g. Excell or draw) Randomization processes occurred by lots in closed envelopes and were performed by a blinded operator

please standardize the font size. Done

Round 2

Reviewer 1 Report

The length of the abstract is still long and does not fit the format suggested by the journal.

In addition, the authors answered that the surgical guide for gingival support was accurate, but most of the studies said that the accuracy was inferior to the tooth support or bone support guide. The same goes for the results of the studies cited by the authors.

The comparison between the traditional surgery and the digital surgery using a surgical guide is an important factor in itself, and it is basic to evaluate whether the clinician has completed the placement in the desired position. And DSD is a digital evaluation of smiles, a process that is meaningful in itself regardless of all-on-4. I still have a hard time understanding the design of this study.

The discussion part still has an inappropriate structure. Discussion should be a structure in which the results obtained by the authors are interpreted in various aspects and compared with existing studies, and the strengths and limitations of this study, future research directions, and clinical usefulness of the results of this study should be mentioned. However, the discussions written by the authors are merely a list of the knowledge examined.

Author Response

1) The length of the abstract is 248 now. 

2) In the article "https://www.ncbi.nlm.nih.gov/pmc/articles/PMC7141387/" conclusions says: "Computer-aided surgery with mucosal-supported templates is a predictable procedure for implant placement. Data showed a discrepancy between the actual dental implant position as compared to the virtual plan, but this was not statistically significant".
In the article "https://pubmed.ncbi.nlm.nih.gov/27062555/", same conclusion: "The mucosa-supported guides indicated a statistically significant greater reduction in angle deviation (P = 0.02), deviation at the entry point (P = 0.002), and deviation at the apex (P = 0.04) when compared to the bone-supported guides. Between the mucosa- and tooth-supported guides, there were no statistically significant differences for any of the outcome measures."

3) The DSD 2d project was matched with che scans of the edentulous model in the CAD software. We used this to obtain the 3d design of the prosthesis.
After that, we matched this 3d project with the DICOM data of the patient's CBCT, using the RealGuide Implant Design software.

4)

The present clinical trial has some limitations, the main one is the follow-up. This type of studies  would need longer follow-up, and our goal is to follow the patients over time, reaching at least 10 years.

One of the purposes of the present study was to suggest the standardization of the aesthetic dental design using a digital protocol.

Reviewer 2 Report

FONT: the text should be justified

INTRODUCTION: page 2, line 58: "from" instead of "form"

MATERIALS AND METHODS:

Can you add the Ethical approval number?

page 3, line 113, and page 4, lines 153-154: can you clarify here the VRS assessment scale (page 11, lines 306-307)?

page 3, lines 124-127: the range of the heights of tilted implants has not been specified yet.

REESULTS:

page 10. line 280: "Two" instead of "2"

page 10, line 301: "at different" instead of "in the different"

DISCUSSION:

A paragraph with strenghts and limitations of the study is missing. As a suggestion, the presence of a scale with only "positive " temrs can not provide the chance to give a negative feedback for the patients.

As a suggestion you can read and cite the manuscript below:

Staderini E, De Luca M, Candida E, Rizzo MI, Rajabtork Zadeh O, Bucci D, Zama M, Lajolo C, Cordaro M, Gallenzi P. Lay People Esthetic Evaluation of Primary Surgical Repair on Three-Dimensional Images of Cleft Lip and Palate Patients. Medicina (Kaunas). 2019 Sep 8;55(9):576. doi: 10.3390/medicina55090576. PMID: 31500380; PMCID: PMC6780772.

REFERENCES:

Please check the number of self-references

Author Response

Ethical board approval number: CE/INT/10/2015.

I did all the other changes requested.

"A paragraph with strenghts and limitations of the study is missing. As a suggestion, the presence of a scale with only "positive " temrs can not provide the chance to give a negative feedback for the patients."

There was a translation error. With "neutral opinion" we meant "ineffective".